# Skill-Learning Method of Dual Peg-in-Hole Compliance Assembly for Micro-Device

**DOI:** 10.3390/s23208579

**Published:** 2023-10-19

**Authors:** Yuting Wu, Juan Zhang, Yi Yang, Wenrong Wu, Kai Du

**Affiliations:** Research Center of Laser Fusion, China Academy of Engineering Physics, Mianyang 621900, China; ytwu_livie@163.com (Y.W.); yangyi_0023@sina.com (Y.Y.); rongwwr@163.com (W.W.); icf802@163.com (K.D.)

**Keywords:** skill learning, peg-in-hole assembly task, reinforcement learning, deep deterministic policy gradient (DDPG), micro-device

## Abstract

For the dual peg-in-hole compliance assembly task of upper and lower double-hole structural micro-devices, a skill-learning method is proposed. This method combines offline training in a simulation space and online training in a realistic space. In this paper, a dual peg-in-hole model is built according to the results of a force analysis, and contact-point searching methods are provided for calculating the contact force. Then, a skill-learning framework is built based on deep reinforcement learning. Both expert action and incremental action are used in training, and a reward system considers both efficiency and safety; additionally, a dynamic exploration method is provided to improve the training efficiency. In addition, based on experimental data, an online training method is used to optimize the skill-learning model continuously so that the error caused by the deviation in the offline training data from reality can be reduced. The final experiments demonstrate that the method can effectively reduce the contact force while assembling, improve the efficiency and reduce the impact of the change in position and orientation.

## 1. Introduction

With the development of a Micro-Electro-Mechanical System (MEMS), the demands for micro-devices are gradually increasing in both type and quality. In order to ensure accuracy and quality, micro-assembly technology is often used to control the assembly tasks.

To reduce the contact force while assembling, two strategies are commonly used: active compliance and passive compliance. A passive compliance strategy mainly produces natural compliance through some auxiliary constructions for devices to adapt to the environment, such as Remote Center Compliance (RCC), Spatial Remote Center Compliance (SRCC), Variable Remote Center Compliance (VRCC), etc. Huang H. [1] focused on the passive compliance tools and designed an adaptive path-planning approach for the robotic grinding and polishing system. Koco E. [2] presented a new hybrid compliance control system for an electrically powered quadruped robot leg, including both active and variable passive compliance parts, by using passive compliance for the filtering of sudden impacts during locomotion.

However, the passive compliance strategy cannot solve the contradiction between rigid parts and flexible insertion demands, the application range is restricted and these kinds of devices often face the problems of poor adaptability, high pertinence and a low success rate of assembly. Therefore, researchers have conducted a lot of research on active compliance strategies. Since the 1970s, with the development of sensors and control technology, force feedback control has developed into one of the main directions of robot-assembly research. The active compliance strategy contains various control methods such as impedance control, force/position hybrid control and learning-based compliance control. Impedance control achieves compliance through the position–velocity–force relationship model of the actuator at the end of the manipulator. It adjusts the robot’s action according to the force feedback. Jin Y. [3] proposed an active compliance control method for the aircraft-cleaning arm, aiming at controlling the contact force in aircraft-skin cleaning by the aircraft-cleaning arm. This method is based on impedance control combined with the adaptive control method to improve the accuracy of the tracking position. And through the fuzzy controller, the impedance parameters are adjusted in real time according to the force error, which improves the accuracy and stability of the task. However, classical impedance and admittance control are environment-dependent, and the force-tracking accuracy cannot meet the requirements [4]. Force/position hybrid control can improve the control accuracy by combining position and force information. Binglong W. [5] proposed a force/position-hybrid-control-based precision peg-in-hole assembly method for industrial robots and designed a new force/position hybrid control strategy based on a servo velocity loop, which can better track high-frequency signals and has a better force-tracking performance than traditional force-control methods. Nevertheless, this kind of method has low efficiency, low accuracy and a slow response for unstructured environments or non-linear assembly tasks.

To solve the problem of compliant assembly and improve the adaptive control performance of the system, considering the uncertainty of the system, including modeling errors, structural uncertainties and other unfavorable conditions [6], researchers have also conducted some research on adaptive systems. Qin X. [7] proposed a load-adaptive control system of Marine hydraulic steering gear. The system adopted a three-closed-loop control structure, using steering gear for data acquisition, and ran a fuzzy PID adaptive control algorithm to output the multi-level controlled quantity. This method enables the system to control the rudder of the Angle of Marine hydraulic steering gear according to the load change and accurately track the load sine signal. With the development of artificial intelligence, researchers have also combined deep learning methods with adaptive systems. Hua O.Y. [8] designed an adaptive PID controller based on the neural network for adjusting internal parameters, which solves the problem of pre-setting PID parameters through manual experience. This system can reduce the dependence on the operator’s experience and avoid problems such as error increasing and the extension of response time caused by the changes in the environment. Xin L. [9] proposed an adaptive PID ship-motion controller based on a neural network, which is composed of a PID controller, neural network unit and ship-rudder-Angle control unit. By using the adaptive ability of the RBF neural network, accurate and fast control of the ship-rudder Angle is realized. Patino H.D. and Liu D. [10] implemented a neural network to address the classical problems in the Model Reference Adaptive System (MRAS), especially for the first-order continuous-time non-linear dynamical system. They designed a controller structure that can employ a radial basis function network or a feed-forward neural network to deal with the non-linearities of the system, ensuring the system can successfully assure the convergence of the control error around zero.

Reinforcement learning, as one of the critical technologies of artificial intelligence, is widely used in control systems because of its excellent ability to explore the environment and achieve adaptive and compliant control. And, a control system always contains the vision module, the force-perception module, the motion module and the grasping module so that the real-time status information can be obtained. Industrial robots can use reinforcement learning methods to autonomously learn optimal moving actions for the peg-in-hole assembly tasks to adapt to complex and variable environments and requirements. Senda K. [11] proposed a reinforcement-learning-based skill-generating method and added perturbations to the object poses for training to improve self-adaptation for uncertain environments. Zou P. [12] designed a variable impedance controller by combining fuzzy theory and Q-learning to ensure the safety, stability and efficiency of the assembly task by a force controller and the learning-based optimization algorithm. However, the method requires many fuzzy rules and is very complex to implement. The neural network can transform high-dimensional data into low-dimensional features by adjusting the internal nodes and has the characteristics of multiprogramming and nonlinear operations, showing a strong numeration power, adaptability, fault-tolerant ability and self-organizing ability. Therefore, researchers have combined neural networks with reinforcement learning and proposed deep reinforcement learning methods for assembly tasks. Beltran-Hernandez C. C. [13] proposed a reinforcement learning and adaptive control method for peg-in-hole assembly to solve the problem of uncertain poses of holes, using the robot state as the input and outputting action through neural networks, such as the trajectory and adaptive parameters, which improves the efficiency of insertion. Feng S. [14] proposed a variable-parameter impedance-control algorithm by reinforcement learning; the controller parameters are optimized by the deep deterministic policy gradient (DDPG) algorithm, and the fuzzy reward is used to avoid getting into the local optimum dilemma. Xu J. [15] proposed a feedback-exploration strategy based on the DDPG algorithm, and a simple P-controller is used to actuate learning. The approach ensures that the assembly task can be completed without analyzing the contact state, but the stability and effectiveness still need to be improved.

Weak micro-devices are small in size and request high accuracy and are more prone to deformation, especially in interference-fit assembly tasks where tiny deviations in pre-registration can lead to deformation and damage during part insertion. Therefore, for such assembly tasks, it is often necessary to continuously adjust the inserting action according to the status information to reduce the contact force and improve the assembly accuracy.

In this paper, a skill-learning compliant inserting method for weak micro-devices based on simulation pre-training together with online learning is proposed, mainly taking the micro-target assembly problem in the Inertial Confinement Fusion (ICF) experiment as an example. Moreover, the proposed method builds a dual peg-in-hole simulation model of the upper and lower double-hole structure according to the force analysis results and pre-trains it. Then, the trained model is used for online learning of the assembly task. We utilized a DDPG algorithm to learn the assembly skills by setting the positive and negative rewards and added noise to both the state and action to improve the environmental adaptability and exploration efficiency of the model. The simulation space pre-training combined with the online training method effectively improves the training efficiency. The feasibility of the method is verified through experiments. Also, the stability, efficiency and safety of the method in the micro-device assembly task are demonstrated through comparative experiments with the classical assembly method.

## 2. Problem Description and Skill-Learning Framework Design

### 2.1. Dual Peg-in-Hole Task Description of Upper and Lower Double-Hole Structure

A double-hole part is a typical micro-part in the ICF micro-target composition which has two holes on the top and bottom and is usually a thin-wall metal cavity part. The double-hole part needs to be assembled with the metal shaft, and the insertion depth needs to be kept within limits. The assembly process is shown in Figure 1.

The part is internally hollow, and two holes are punched along the radial direction in the middle position. Since the internal structure of the part can not be observed by microscopic vision, it is impossible to guarantee the alignment attitude before the lower hole is loaded, which corresponds to a definite uncertainty in the internal structure. The part is clamped by vacuum adsorption during the loading, which leads to the clamping deflection if the force during the loading process is extensive, corresponding to a definite uncertainty in the part’s degree of freedom. And because of the thin-wall cavity structure of the double-hole part where the thickness of the wall is usually in the order of 30–50 μm, it is very easy to deform. So, the vacuum-adsorption gripper is used to position it, but it is limited in its binding effect, which can lead to errors in position and orientation when the contact force is strong. Additionally, the clearance required is usually in micrometers. The parts’ batch error can lead to the fit mode from clearance to micro-clearance or even micro-interference. In addition, the mechanical arm is widely used to operate micro-parts, which meets the needs of operating parts in a large space, but the uncertainty of the assembly environment is increased by the instability of the degree of freedom of the multi-joint tandem mechanical arm. Therefore, for the compliance assembly of weak micro-parts in this kind of uncertain environment, the reinforcement learning method is an effective approach. The ability to self-adapt and continuously optimize the loading strategy through reinforcement learning can improve the loading success rate and reduce the impact of excessive contact force.

### 2.2. Skill-Learning Framework for Dual Peg-in-Hole Compliance Assembly

The following challenges still remain while applying the reinforcement learning methods for online learning directly:The binding force is limited, so the position and orientation are unstable.For weak micro-parts, it is costly to train online directly because the deformation can occur when the contact force is too strong.The control precision is limited due to the instability of the multi-joint robot arm.The robot arms may collide during assembly.

To satisfy the actual needs, we established a network for skill learning combining mechanical-force-analysis-based offline training and experiment-based online training. To improve the training efficiency, expert action based on the P-controller and incremental action output by the network are added for training; this achieves the purpose of compliance assembly, and we adjust the insert action according to the force information.

The framework of skill learning is illustrated in Figure 2. Based on the force analysis of the dual peg-in-hole assembly task of the upper and lower double-hole structure to establish the simulation model for offline training, we add noise to obtain the data of the insertion task to avoid falling into the local minima and use the deep reinforcement learning network to train the intelligent agent. The online learning strategy is based on the feedback of multidimensional force/moment information in real space, and it constantly updates the data in the experience pool for agent optimization.

## 3. Simulation Model Modeling

Because the network training needs many samples and has easy-to-occur characteristics of deformation, it is easy to cause damage to parts and robots when the operation is not appropriate. Therefore, in order to improve the sample acquisition efficiency and reduce the collision risk and the wastage of components, this section establishes a simulation model by analyzing the contact force of the dual peg-in-hole assembly process, which is used to obtain the pre-training data. The simulation training takes the force information and the number of action steps as the basis for judging success. Whenever the force exceeds the threshold range or the number of execution steps exceeds the threshold range, the assembly will be judged as a failure and the simulation model will be initialized to return to the initial state and start a new round of action.

### 3.1. Force Analysis of the Assembly Process

In order to analyze the contact force of different states in the assembly process, a mechanical model is established, and a rectangular coordinate system is set up with the center of the very bottom circle *O* as the coordinate origin, as shown in Figure 3a. According to the symmetry, the spatial assembly model can be simplified to a two-dimensional planar dual peg-in-hole assembly model of upper and lower double-hole structures. We assume that the pegs and holes are point contacts, the pegs are strictly rigid and the holes are elastic. The point with the most considerable deformation is considered as the contact point for analysis.

The state before assembly is shown in Figure 3b; the shaft mainly contains the top characteristic circle $B_{1}$ and bottom characteristic circle $B_{2}$, and the hole contains four characteristic circles $A_{1}$, $A_{2}$, $A_{3}$ and $A_{4}$, where *D* is the hole diameter, *d* is the shaft diameter, $l_{z}$ denotes the Z-directional deviation after alignment and θ denotes the angular deviation. $l_{1}$ is the shaft length, $l_{2}$ is the wall thickness of the hole and $l_{b}$ is the inner-space length.

During the assembly process, there are four phases according to the mandrel inserting depth: the upper-hole approaching phase, upper-hole inserting phase, lower-hole approaching phase and lower-hole inserting phase. Since the upper-hole approaching phase is not subject to contact forces, only the last three phases are taken into consideration.

As shown in Figure 4, the assembly process contains a total of nine contact states. According to the number of contact points, the contact states can be classified into single-point contact, double-point contact and three-point contact. And all contact points can be divided into hole-based contact points and shaft-based contact points, where the main feature of the hole-based contact points is that the contact points are located in the characteristic circle of the shaft, and the main feature of the shaft-based contact points is that the contact points are located in the characteristic circle of the hole. As the inserting depth increases, the contact point position changes, and a different characteristic circle needs to be selected to calculate the contact force.

The contact point shown in Figure 5 is the shaft-based contact point. The characteristic circle at the bottom of the shaft is the reference circle. According to $F_{N}=E\delta$, $F_{f}=\mu F_{N}$, the contact force at the point ($x_{1}$,$y_{1}$,$z_{1}$) can be calculated by:(1)$$\left\{\begin{array}{c} F_{x}^{1}=\frac{y_{1}}{\sqrt{y_{1}^{2}+x_{1}^{2}}}\cdot F_{Nxy}^{1}\\ F_{y}^{1}=\frac{x_{1}}{\sqrt{y_{1}^{2}+x_{1}^{2}}}\cdot F_{Nxy}^{1}\\ F_{z}^{1}=F_{f}^{1}+F_{Nz}^{1}=\mu F_{Nxy}^{1}+F_{Nz}^{1} \end{array}\right.$$
where *E* is the Elastic Modulus, δ is the deformation perpendicular to the contact surface and μ is the friction coefficient which can be obtained by pre-calibration. $F_{Nxy}^{1}$ and $F_{Nz}^{1}$ are calculated from the radial deformation $\Delta_{xy}$ and axial deformation $\Delta_{z}^{1}$, respectively.

Similarly, Figure 6 shows that the contact point is based on the hole, and the characteristic circle at the bottom of the hole is the reference circle. The analysis of point 2 can be obtained:(2)$$\left\{\begin{array}{c} F_{x}^{2}=\frac{y_{2}}{\sqrt{y_{2}^{2}+x_{2}^{2}}}\cdot F_{Nxy}^{2}\\ F_{y}^{2}=\frac{x_{2}}{\sqrt{y_{2}^{2}+x_{2}^{2}}}\cdot F_{Nxy}^{2}\\ F_{z}^{2}=F_{f}^{2}=\mu F_{Nxy}^{2} \end{array}\right.$$

In summary, the assembly process contains a total of nine contact states and two types of contact points, and the total contact force $F=[F_{x},F_{y},F_{z}]$ can be calculated by choosing different calculation methods according to the contact-point types.

### 3.2. Training-Data-Acquisition Strategy in the Simulation Environment

According to the force analysis results, theoffline training simulation model was built based on Python. Theconstruction of the simulation environment takes the edge circles of the end faces of the parts as the main calculation object and discretizes the circle as points for searching the contact point. Duringthe training process, assoon as the action information is generated according to the current force situation, themodel will calculate and update the force information according to the contact-point situation; return to the network for the next step; andstore the action, force information and the reward of the current step for training.

The contact-force calculation method is shown in Figure 7, which can be divided into the initial pose-generating phase, contact-point searching phase and contact-force calculating phase. To facilitate the calculation, the characteristic circles of the end faces are used to replace the complex side surface for calculation.

During the initial pose-generating phase, *N* groups of position and orientation errors are randomly generated to simulate the errors after alignment in the real assembly process, and the initial pose of the mandrel can be calculated:(3)$$\left(\begin{array}{cccc} x_{i}^{\prime } & y_{i}^{\prime } & z_{i}^{\prime } & 1 \end{array}\right)=\left(\begin{array}{cccc} x_{i} & y_{i} & z_{i} & 1 \end{array}\right)\cdot \;R_{x}\cdot \;R_{y}\cdot \;R_{z}\cdot \;T_{r}$$
where ($x_{i}$,$y_{i}$,$z_{i}$) is the coordinate on the characteristic circle without deviation; ($x_{i}$,$y_{i}$,$z_{i}$) is the coordinate on the characteristic circle after initial pose generating; and $R_{x}$, $R_{y}$ and $R_{z}$ are the rotation matrices around the X, Y and Z axes, respectively. $T_{r}$ is the translation matrix.

Subsequently, based on the analysis of the contact-point type, two searching methods can be obtained: the shaft-based searching method and the hole-based searching method. Take the upper-hole inserting phase as an example: the shaft-based searching method is mainly used for searching shaft-based contact point 1 shown in Figure 5. The center of the characteristic circle of shaft $B_{2}$ can be considered as the reference point. The elliptic curve projected from the characteristic circle at the bottom of the hole to the end plane of the shaft can be considered as the searching curve, searching for the point with minimum $\Delta_{xy}^{k}$, whereby *k* is the index of the point:(4)$$\Delta_{xy}^{k}=\sqrt{{\left(x_{p}-x_{b}\right)}^{2}+{\left(y_{p}-y_{b}\right)}^{2}}-R_{b}$$
where ($x_{b}$,$y_{b}$,$z_{b}$) is the coordinate of the reference point and ($x_{p}$,$y_{p}$,$z_{p}$) is the coordinate of the point on the searching curve. $R_{b}$ is the radius of the reference circle. Similarly, the hole-based search method is mainly used to search the hole-based contact point 2 shown in Figure 6. In the upper-hole inserting phase, the center of the characteristic circle of the hole is taken as the reference center, and the elliptic curve projected from the characteristic circle of the shaft on the end face of the hole can be considered as the searching curve, searching for the point with maximal $\Delta_{xy}^{k}$. If there is no such point that satisfies the equation $\Delta_{xy}^{k}\leq$ 0 while using the shaft-based search method or satisfies $\Delta_{xy}^{k}\geq$ 0 while using the hole-based search method, it is deemed that there is no contact between the two parts. In the contact-force calculating phase, the axial force and radial forces can be obtained by Equation (1) or (2) mentioned in Section 2.1. Finally, the current state $s_{t}$ can be obtained through the calculation of the contact force, and the incremental action $a_{t}^{i}$ can be obtained by the action network. The inserting action $a_{t}$ can be calculated by the Jacobian matrix of force, and the new state $s_{t+1}$ can be updated. The reward value can be calculated by the current state $s_{t}$, incremental action $a_{t}^{i}$ and new state $s_{t+1}$, and these data can be saved for network training.

## 4. Skill Learning

The DDPG extends DQN into a continuous action space, which can output specific actions corresponding to the maximum long-term benefits and improve stability and convergence. Therefore, the DDPG framework is used in this paper.

### 4.1. Network Based on DDPG

The DDPG network in this paper is built based on the AC framework, as shown in Figure 8. The action network is used to obtain the incremental action, and the critic network can output the action value to update the actor network and optimize the inserting action. Both the action network and the critic network are whole conjunction neural networks. The inserting action $a_{t}$ contains expert actions $a_{t}^{e}$ and incremental action $a_{t}^{i}$ output by the action network:(5)$$a_{t}=a_{t}^{e}+a_{t}^{i}$$

The expert action $a_{t}^{e}={\left[d_{xt}^{e},d_{yt}^{e},d_{zt}^{e}\right]}^{T}$ is used to ensure the safety and success of the insertion and is obtained by the P-controller:(6)$$\left[\begin{array}{c} d_{xt}^{e}\\ d_{yt}^{e}\\ d_{zt}^{e} \end{array}\right]=-\lambda J_{F}\left[\begin{array}{c} f_{xt}\\ f_{yt}\\ 0 \end{array}\right]+\left[\begin{array}{c} 0\\ 0\\ \delta_{z} \end{array}\right]$$
where $J_{F}$ is the Jacobi matrix of force and $f_{xt}$ and $f_{yt}$ are the contact-force components along the $X_{F}$ and $Y_{F}$, respectively. $d_{xt}^{e}$, $d_{yt}^{e}$ and $d_{zt}^{e}$ are the components of the expert action along the $X_{M}$, $Y_{M}$ and $Z_{M}$, respectively. A small $\delta_{z}$ is set to ensure the insertion safety. Constant λ∈[0,1].

The incremental action is obtained from the actor network: $a_{t}^{i}=\mu \left(s_{t}|\theta^{\mu }\right)$, where $s_{t}$ is the state at *t* moment. According to the chain rule, the parameters $\theta^{\mu }$ of the actor network can be updated by computing the policy gradient:(7)$$\nabla_{\theta^{\mu }}J\left(\theta^{\mu }\right)=E\left[\nabla_{a}Q\left(s_{t},a_{t}^{i}|\theta^{Q}\right)\cdot \nabla_{\theta^{\mu }}\mu \left(s_{t}|\theta^{\mu }\right)\right]$$

The critic network takes the state $s_{t}$ and incremental action $a_{t}^{i}$ as inputs, outputs the action values $Q(s_{t},a_{t}^{i}|\theta^{Q})$ and optimizes by minimizing the loss:(8)$$L\left(\theta^{Q}\right)=E\left[{\left(Q\left(s_{t},a_{t}^{i}|\theta^{Q}\right)-y_{t}\right)}^{2}\right]$$
where $y_{t}=R_{t}+\gamma Q^{\prime }\left(s_{t+1},\mu^{\prime }\left(s_{t+1}\right)|\theta^{Q}\right)$ and γ is the discount factor. The target action network and the target critic network have the same structure as the action network and the critic network. It periodically updates the parameters of the target network:(9)$$\left\{\begin{array}{c} \theta^{Q^{\prime }}=\tau \theta^{Q}+\left(1-\tau \right)\theta^{Q^{\prime }}\\ \theta^{\mu \prime }=\tau \theta^{\mu }+\left(1-\tau \right)\theta^{\mu^{\prime }} \end{array}\right.$$
where the learning rate τ∈(0, 1).

### 4.2. State and Action Definition

During the inserting process, since the inserting action is related to the contact force and the insertion depth at that moment, the state and action at the *t* moment can be defined as:(10)$$\left\{\begin{array}{c} s_{t}={\left[f_{xt},f_{yt},f_{zt},p_{zt}\right]}^{T}\\ a_{t}={\left[d_{xt},d_{yt},d_{zt}\right]}^{T} \end{array}\right.$$

Since the parts’ orientation is guaranteed by the alignment step, the inserting action mainly aims at controlling the position of parts, where, $f_{xt}$, $f_{yt}$ and $f_{zt}$ denote the contact force along $X_{F}$, $Y_{F}$ and $Z_{F}$, respectively. $p_{zt}$ denotes the total insertion depth along the $Z_{M}$. $d_{xt}$, $d_{yt}$ and $d_{zt}$ denote the action along $X_{M}$, $Y_{M}$ and $Z_{M}$ respectively.

### 4.3. Reward

In order to ensure the safety of the assembly, while taking efficiency into account, this paper uses the contact force to calculate the positive reward $R_{pt}$, and the negative reward $R_{nt}$ can be calculated according to the inserting depth:(11)$$R_{t}=R_{pt}-R_{nt}$$

Since the part is limited by the fixture step along $Y_{M}$, the displacement generated by the force is small while the position and orientation along $X_{M}$ rely on the adsorption force, which is more likely to lead to displacement, so the contact forces along X and Y are weighted separately; the larger the contact force, the smaller the reward obtained:(12)$$R_{pt}=1-\alpha_{1}\frac{f_{xt}}{f_{X\max}}-\alpha_{2}\frac{f_{yt}}{f_{Y\max}}$$
where $f_{Xmax}$ and $f_{Ymax}$ are the maximum allowable radial forces along $X_{F}$ and $Y_{F}$, respectively. $\alpha_{1}$ and $\alpha_{2}$ are the contact-force weight factors along $X_{F}$ and $Y_{F}$, respectively, satisfying $\alpha_{1}+\alpha_{2}=1$. The negative reward $R_{nt}$ is calculated by the difference between the actual inserting depth and the ideal inserting depth. The larger the difference, the larger the reward value.

Additionally, too long an insertion depth has more impact than too short an insertion depth in order to reduce wastage. Since the ideal inserting depth is related to the current radial force and axial force, we can obtain:(13)$$R_{nt}=\left\{\begin{array}{c} \frac{d_{zt}-R_{p(t-1)}d_{\max}}{d_{\max}},d_{z}\geq R_{p(t-1)}d_{\max}\\ \frac{R_{p(t-1)}d_{\max}-d_{zt}}{\omega d_{\max}},d_{z}<R_{p(t-1)}d_{\max} \end{array}\right.$$
where $d_{zt}$ is the inserting depth, $d_{max}$ is the maximum inserting depth allowed in the Z-direction, $f_{zt}$ is the axial force in the Z-direction after acting and the scale factor ω>1.

### 4.4. Environment Imitation and Action Optimization

During reinforcement learning training, in order to ensure the efficiency of exploration and reduce the negative effect due to a large number of random explorations, in this paper, environment and action noises are added to train the intelligent body to complete the exploration task efficiently. First, add Gaussian noise to the action $a_{t}$:(14)$$a_{t}^{n}=a_{t}+N\left(0,\sigma_{1}^{2}I\right)$$
where $a_{t}^{n}$ is the action with noise; $\sigma_{1}$ is the standard deviation of the exploration action, which determines the intensity of the exploration; and $N(0,\sigma_{1}^{2}I)$ is the Gaussian noise. Reducing the intensity of exploration when the insertion is successful results in:(15)$$\sigma_{1}=\left\{\begin{array}{c} \sigma_{\min},\;\mathrm{succeed}\;\\ \sigma_{\max},\;\mathrm{failed}\; \end{array}\right.$$

Meanwhile, in order to simulate the effect of environmental changes in real space, noise is added to the state after the execution of the action to improve the model’s adaptive capability:(16)$$s_{t}^{n}=s_{t}+N\left(0,\sigma_{2}^{2}I\right)$$
where $s_{t}^{n}$ is the environment with noise and $\sigma_{2}$ is the standard deviation of the environmental noise. The pseudocode of the algorithm is shown as Algorithm 1.
**Algorithm 1** Algorithm based on DDPG for Dual Peg-in-hole Assembly Task.Initialize parameters $\sigma_{1}\leftarrow 0.1$, $\sigma_{2}\leftarrow 0.2$, $M_{E}$, target network update period *T*, learning rate τ**for** epoch = 1,2, …, **do**    Initialize $s_{0}$    **for** t = 1,2, …, **do**       Calculate expert action $a_{t}^{e}$ and select incremental action $a_{t}^{i}$       Calculate exploratory action $a_{t}^{n}=a_{t}^{e}+a_{t}^{i}+N\left(0,\sigma_{1}^{2}I\right)$       Execute action $a_{t}^{n}$, observe reward $R_{t}$ and observe new state $s_{t+1}$       **if** $s_{t+1}$ is terminal **then**           Stop epoch and reset environment state       **end if**       Store ($s_{t}$,$a_{t}^{n}$,$R_{t}$,$s_{t+1}$) in the replay buffer $M_{E}$       Randomly sample *N* transitions from $M_{E}$       Set $y_{t}=R_{t}+\gamma Q^{\prime }\left(s_{t+1},\mu^{\prime }\left(s_{t+1}\right)|\theta^{Q}\right)$       Update the critic network by minimizing the loss:       $L\left(\theta^{Q}\right)=E\left[{\left(Q\left(s_{t},a_{t}|\theta^{Q}\right)-y_{t}\right)}^{2}\right]$       Update the actor network by using the sampled policy gradient:       $\nabla_{\theta }J\left(\theta^{\mu }\right)=E\left[\nabla_{a}Q\left(s_{t},a_{t}|\theta^{Q}\right)\cdot \nabla_{\theta^{\mu }}\mu \left(s_{t}|\theta^{\mu }\right)\right]$       **if** *t* mod *T* = 0 **then**           Update target networks:     $\left\{\begin{array}{l} \theta^{Q^{\prime }}=\tau \theta^{Q}+\left(1-\tau \right)\theta^{Q^{\prime }}\\ \theta^{\mu^{\prime }}=\tau \theta^{\mu }+\left(1-\tau \right)\theta^{\mu^{\prime }} \end{array}\right.$       **end if**       $s_{t}\leftarrow s_{t+1}$    **end for** $\sigma_{1}=\left\{\begin{array}{l} \sigma_{\min},\;\mathrm{succeed}\\ \sigma_{\max},\;\mathrm{failed}\; \end{array}\right.$ **end for**

## 5. Experiments

The model-driven deep deterministic policy gradient algorithm can improve the efficiency and interpretability of the algorithm. To ensure the training effect and reduce the parts consumption, firstly, the agent is trained in the simulation. And then it optimizes the network by real-space online training.

### 5.1. Simulation Training Setup

According to the results of the contact-force analysis, build the dual peg-in-hole simulation model of the upper and lower double-hole structure with the following parameter settings as shown in Table 1.

In the simulation environment, the bore diameter *D* is 1.01 and the shaft diameter *d* is 1.0. The length $l_{1}$ is 4, the upper and lower bore thicknesses $l_{2}$ are 0.5 and the distance $l_{b}$ between the upper and the lower hole is 1.5. The δ is 50. The single step length ($d_{xt}$, $d_{yt}$, $d_{zt}$) is controlled within the range of [20, 20, 60], respectively; $f_{Xmax}$ is 70 and $f_{Ymax}$ is 100; and the upper-limit value of the axial contact force is 1000. When the contact force exceeds the allowed range or the maximum number of steps exceeds 1000, the assembly task can be diagnosed as a fail.

We randomly generated 300 groups of the initial state with position deviation within 10 μm and orientation deviation within 0.3${}^{\circ }$. The results are shown in Figure 9.

### 5.2. Online Training Setup

For online training, the assembly object is a decussation target where the shaft material is aluminum with a diameter *d* of 1 mm and $l_{1}$ length of 4 mm, the cavity material is gold with a hole diameter *D* of 1.01 mm, the distance $l_{b}$ between the upper and lower holes is 2.5 mm, the wall thickness $l_{2}$ is 0.5 mm and the gold cavity is fixed by the gripper beyond the force sensor.

The assembly experiment platform mainly includes a six DOF industrial robot and a six DOF operating platform. The detection module includes a variable magnification vision system, vertical servo vision system and micro-force detection system. The variable magnification vision system mainly consists of two horizontal micro-vision channels and is installed on the three DOF motion platform. The vertical servo vision and the gripper are fixed at the end of the robot. The micro-force detection module collects force information by a force sensor.

To improve the training effect, the network parameters are further optimized by online training considering the environment information of the real assembly task, the insertion data are stored and the network is periodically updated.

The online learning method exchanges the state and action information through socket communication technology and continuously stores the data of each actual insertion in the experience replay of size 200 in the form of ($s_{t}$,$a_{t}^{i}$,$R_{t}$,$s_{\left(t+1\right)}$), and it updates the network weights. To ensure the assembly safety, the operation station motion speed is limited to 5 mm/s. To a certain extent, it can reduce the impact of the error between the simulation environment and the actual insertion environment and builds a bridge between the intelligent body and the real assembly environment.

### 5.3. Assembly Experiment Results Based on the Proposed Skill-Learning Method

Before each insertion, the alignment between two parts is performed by using visual inspection, including the rough alignment based on the robot arm and the fine alignment based on the operation stage, with the final position deviation controlled at 2 μm and the orientation deviation controlled at 0.1${}^{\circ }$.

Due to the upper and lower double-hole structure, there are two sections with large contact forces while inserting, as shown in Figure 10. The maximum force along $X_{F}$ is controlled within 100 mN, the maximum force along $Y_{F}$ is controlled within 40 mN, the maximum axial force is controlled within 90 mN and the insert step is 53.

Based on the trained model, 30 inserting experiments were carried out and the model was continuously optimized by online training, and the experimental results are shown in Figure 11. The forces are mainly concentrated in the [0, 44] mN interval, meaning they cannot easily cause the deformation of weak micro-parts and the deflection of posture, and they have a good assembly effect.

### 5.4. Comparative Experiment Results and Analysis

Since the assembly is a static process, the conventional P-controller was used for comparison experiments, setting the gain to 800 μm/N, the Z-direction single-step length to 60 μm, the total loading stroke to 3200 μm and the finish axial force to 1000 mN. This method is more dependent on the human-set parameters, and the Z-direction step length cannot be adaptively adjusted so that force oscillation may occur during the insertion process, and the three-point contact situation, as shown in Figure 4i, may happen.

Comparing the results of the traditional P-control and the trained model based on a combination of offline and online training for the dual peg-in-hole assembly task, as shown in Figure 12, it can be found that the two methods have similar contact forces during the upper-hole inserting phase, mainly because the fine alignment makes the initial position and orientation deviation smaller, while the traditional method tends to have a much larger contact force during the lower-hole inserting than the reinforcement learning inserting method, on account of the deviation caused by the unavailability of the correct contact force during the lower-hole approaching phase while using the P-controller. Meanwhile, insertion using the reinforcement learning method can adjust the step length along Z adaptively according to the force information and reduce the damage of part deformation. Therefore, for the dual peg-in-hole task of the upper and lower double-hole structure, the combination of the offline and online reinforcement learning method can reduce the contact force and avoid the failure of assembly due to the part posture deviation. In summary, the method proposed in this paper can perform a dual peg-in-hole assembly task of upper and lower double-hole structures well with a higher assembly success rate and less contact force, improves the assembly safety, reduces the hidden danger of the deformation of weak micro-parts by contact force and requires less actual samples for pre-training in the simulation environment, which improves the training efficiency and reduces the cost.

## 6. Conclusions

For the compliance assembly tasks of micro-devices, to avoid the impact on the assembly results due to the limited positional constraint force, part deformation and instable degrees of freedom of the multi-joint robot arm, a skill-learning model for the dual peg-in-hole task of upper and lower double-hole structures based on the combination of offline training in a simulation space and online learning in a real space is proposed. The method is based on the AC framework, using the DDPG algorithm and designing positive and negative rewards for the agent to train according to the part characteristics, adding action and environmental noise to simulate the assembly environment and improving the exploration efficiency. In order to improve the algorithm interpretability and the efficiency of training data acquisition, a simulation model is built to generate assembly data for pre-training. Then, the model is continuously optimized by online training methods. The effectiveness and advantages of the proposed method were demonstrated through comparison experiments, and the results show that the method proposed in this paper can finish the assembly task with less contact force and has a strong self-adaptive capability for the environment, which is essential for improving the safety and efficiency of the peg-in-hole assembly task of micro-devices. Meanwhile, for other peg-in-hole assembly tasks with complicated structures, the model can be adapted to more assembly tasks by adjusting the simulation model. Subsequently, model matching and selecting can be integrated to complete the skill transfer of multiple assembly tasks.

## Figures and Tables

**Figure 1 sensors-23-08579-f001:**
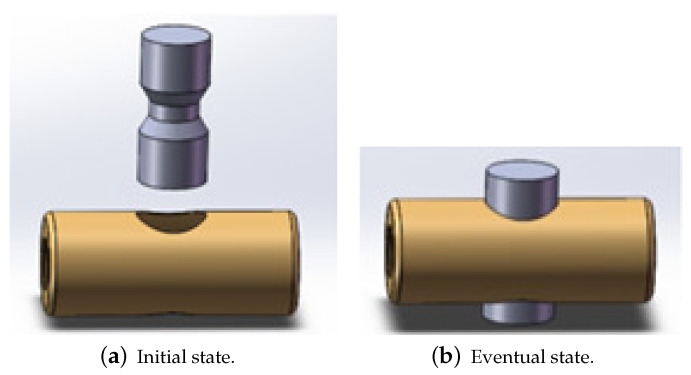
Diagrams of the assembly process.

**Figure 2 sensors-23-08579-f002:**
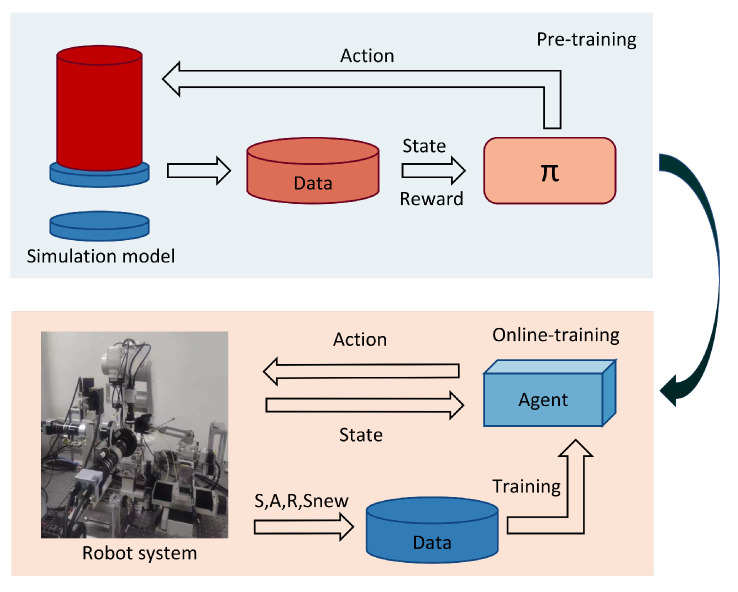
This is the framework of skill learning.

**Figure 3 sensors-23-08579-f003:**
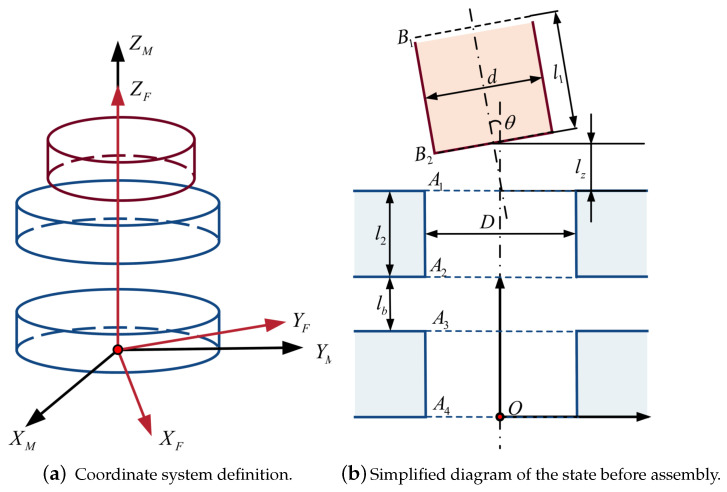
Basic information of force analysis model.

**Figure 4 sensors-23-08579-f004:**
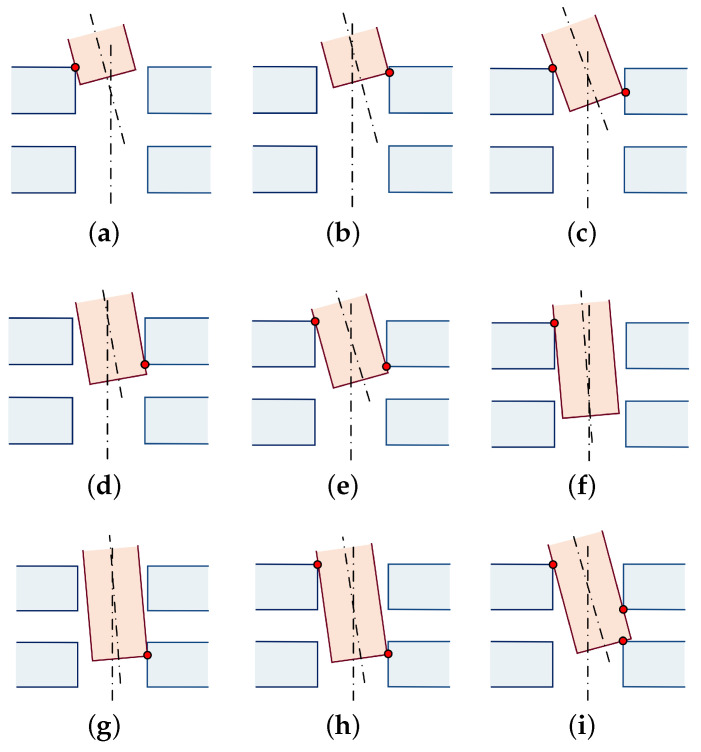
Results of assembly phase analysis: (**a**–**c**) upper-hole inserting phase; (**d**,**e**) lower-hole approaching phase; and (**f**–**i**) lower-hole inserting phase.

**Figure 5 sensors-23-08579-f005:**
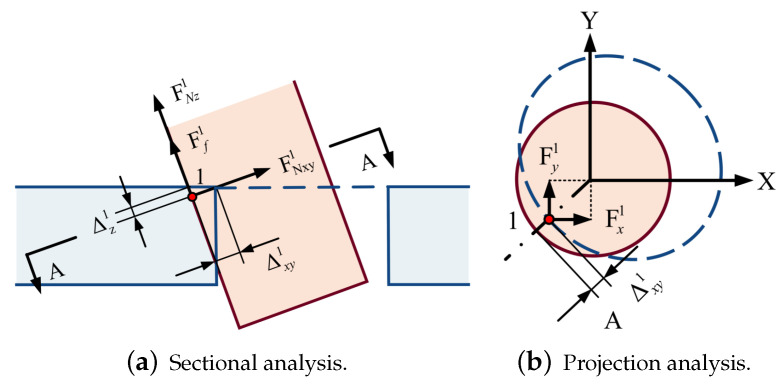
Shaft-based contact-point force analysis.

**Figure 6 sensors-23-08579-f006:**
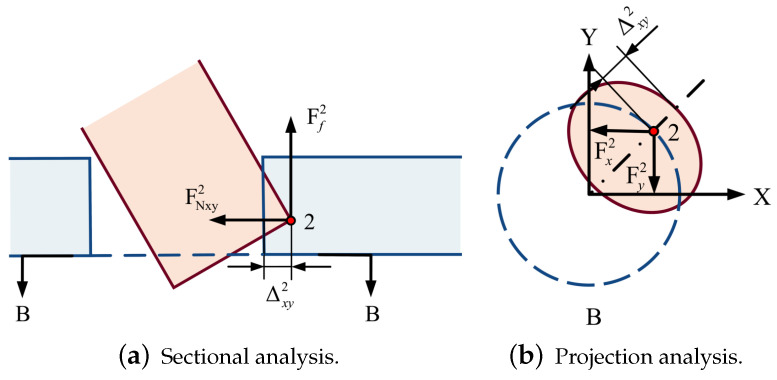
Hole-based contact-point force analysis.

**Figure 7 sensors-23-08579-f007:**
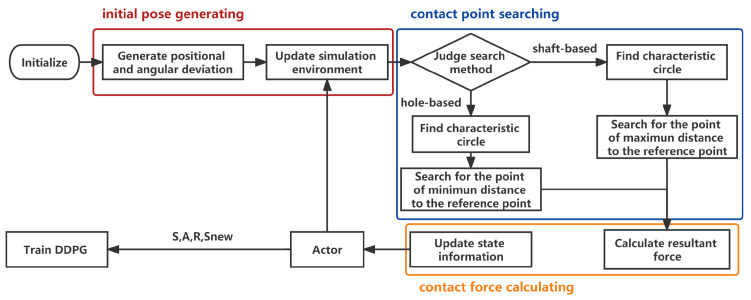
Data acquisition flow chart from simulation model.

**Figure 8 sensors-23-08579-f008:**
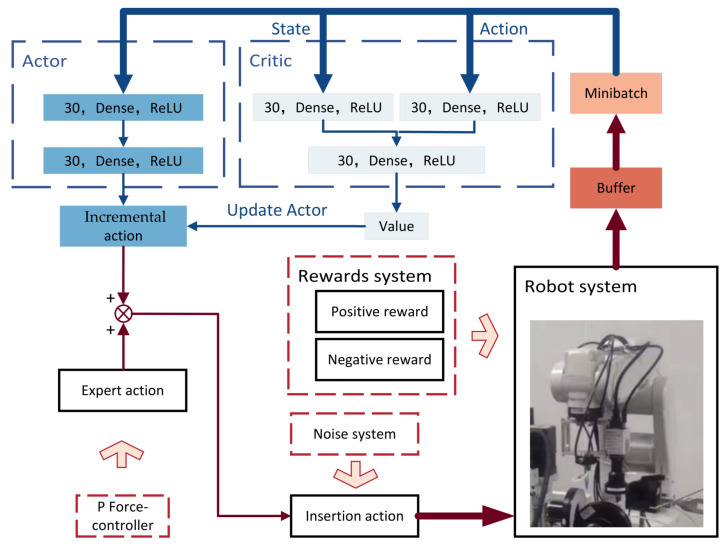
The framework of DDPG algorithm.

**Figure 9 sensors-23-08579-f009:**
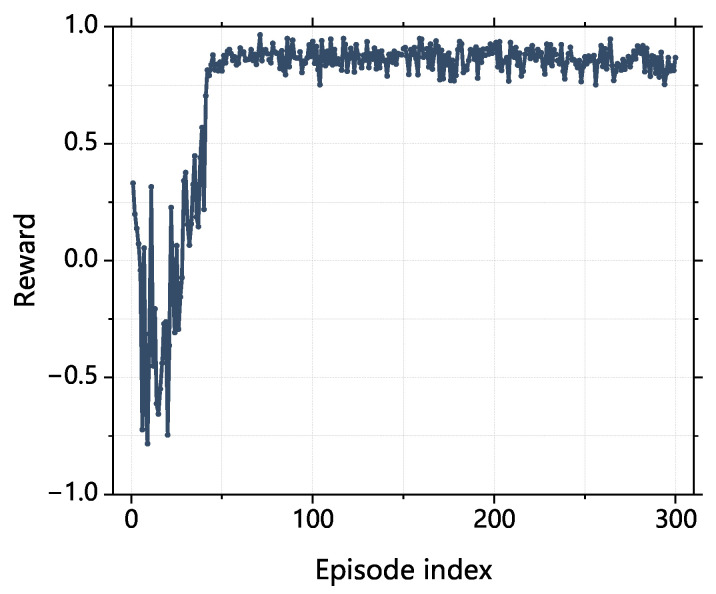
Simulation training result.

**Figure 10 sensors-23-08579-f010:**
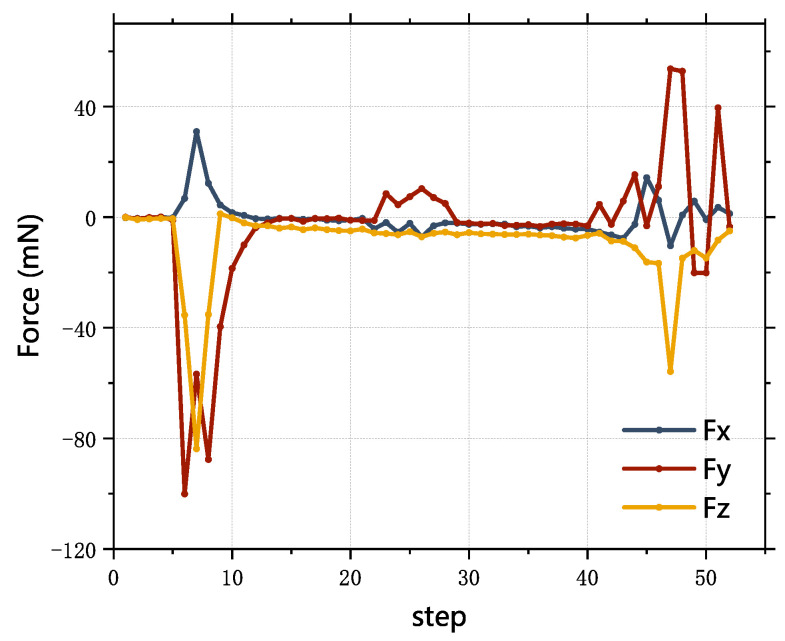
Contact force while inserting.

**Figure 11 sensors-23-08579-f011:**
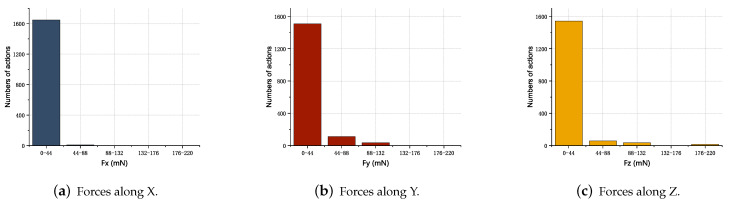
Statistical diagram of 30 groups of experimental forces.

**Figure 12 sensors-23-08579-f012:**
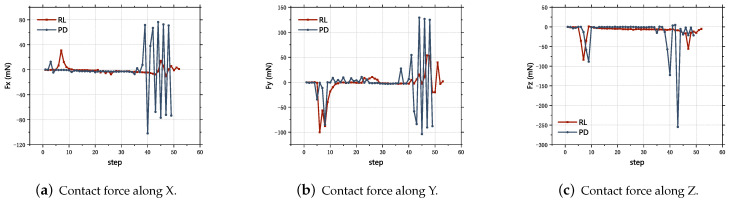
Contact force comparison.

**Table 1 sensors-23-08579-t001:** Training parameters.

Parameters 1	Value	Parameters 3	Value
$\sigma_{max}$,$\sigma_{min}$	0.1,0.4	Experience replay size $M_{E}$	200
$\alpha_{1}$,$\alpha_{2}$	0.7,0.3	Learning rate τ	0.001
ω	2	Batch size *N*	32
γ	0.99	Self-learning rounds	200
λ	0.2	Network update period *T*	0.1

## Data Availability

Not applicable.

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
