# Peer review of "Skill-Learning Method of Dual Peg-in-Hole Compliance Assembly for Micro-Device"

_sensors, 2023, doi:10.3390/s23208579_

Round 1

Reviewer 1 Report

>The acronym DDPG should be previously explained in the text (keywords).

>The text should be improved (line 22).

>The introduction should contain additional information about adaptive control systems based on neural networks (it is related to the scope of the work). Moreover, the hardware aspects need to be analyzed shortly.

>I suppose the dot is necessary (line 130).

>How was the friction coefficient identified (equation 1)?

>Could you provide following analysis related to the influence of the discount factor on the final results (equation 8)?

>How was the learning rate selected (equation 9)?

>Details of the simulation model implementation are not presented.

>Have you considered the experiment for the control system?

It is acceptable, minor improvements of the English grammar are required.

Author Response

We tried our best to improve the manuscript and made some change. We appreciate for your warm work earnestly, and hope the correction will meet with approval. Please see the attachment.

Reviewer 2 Report

The paper addresses an interesting topic (compliant assemblies) and proposes a modern solution method (using the analytical model to train a machine learning algorithm). However, I believe that two important clarifications are necessary: 1. why the method is presented on a particular example (assembly of the metal shaft with double hole part) ? The simple statement that it is a common situation is not enough. 2. what exactly will change in the method if the assembly is different from the example chosen for the presentation?

Author Response

(The authors gave the same response as above.)

Reviewer 3 Report

The paper presents reinforcement learning method for dual peg-in-hole assembly for micro-devices, which combines offline simulation and online realistic sensing data as re-training analysis. The works are very interesting and meaningful to solve the actual engineering issues. But there are many problems that can be improved as follows:

1. In Section 2, the description of dual peg-in-hole task has a little confusing for the specific micro-assembly problems, such as the unstable multi-joint degree of freedom, uncertain assembly environment, etc. I suggest that the dual peg-in-hole assembly task should be specified how to combine reinforcement learning method for the specific scientific problems. I think you can explain clearer.

2. The simulation model is very fantastic to acquire the simulation contact force, but there is a big question how to link the updating state information? And you need to give more explanations about the simulation environment updating and actor role.

3. I don’t know what your data is from? Simulation or sensing collection? If your data is collected by sensing devices, you should give some explanations. If not, I suggest that you can combine them to make your proposed model more novelty.

4. I strongly recommend that more related papers should be cited in your paper.

In a word, the paper is good to present a novel method to solve the precise assembly. I suggest that the paper can be accepted after many improvements.

The language expression is fine.

Author Response

(The authors gave the same response as above.)
